# Outward-oriented sites within clustered CTCF boundaries are key for intra-TAD chromatin interactions and gene regulation

Xiao Ge[1,2,4], Haiyan Huang [1,2,4], Keqi Han[1,2], Wangjie Xu[3], Zhaoxia Wang[3] & Qiang Wu [1,2] ✉

CTCF plays an important role in 3D genome organization by adjusting the strength of chromatin insulation at TAD boundaries, where clustered CBS (CTCF-binding site) elements are often arranged in a tandem array with a complex divergent or convergent orientation. Here, using *Pcdh* and *HOXD* loci as a paradigm, we look into the clustered CTCF TAD boundaries and find that, counterintuitively, outward-oriented CBS elements are crucial for inward enhancer-promoter interactions as well as for gene regulation. Specifically, by combinatorial deletions of a series of putative enhancer elements in mice in vivo or CBS elements in cultured cells in vitro, in conjunction with chromosome conformation capture and RNA-seq analyses, we show that deletions of outward-oriented CBS elements weaken the strength of long-distance intra-TAD promoter-enhancer interactions and enhancer activation of target genes. Our data highlight the crucial role of outward-oriented CBS elements within the clustered CTCF TAD boundaries in developmental gene regulation and have interesting implications on the organization principles of clustered CTCF sites within TAD boundaries.

The interphase genome is organized into highly dynamic structures including chromosome territory, compartment, topologically associated domain (TAD), and chromatin loop[1]. CTCF (CCCTC-binding factor) is a key architecture protein for interphase 3D genome organization and CBS (CTCF-binding site) elements throughout the mammalian genomes function as topological insulators to block aberrant enhancer activation and ensure proper gene expression, or to mark the boundaries between euchromatin and heterochromatin[2–4]. A prevailing model for interphase chromosome folding, known as loop extrusion, posits that the cohesin complex extrudes chromatin fibers bidirectionally into expanding loops until blocked asymmetrically by orientated CBS elements[5–7]. Specifically, continuous ATP-driven active loop extrusions lead to gradually larger-sized chromatin loops which

are stabilized by pairs of convergent or opposite CBS elements. This model explains TAD formation and is consistent with the observations that the two flanking boundaries of each TAD are associated with mostly convergent forward-reverse CBS elements[8,9]. However, it is puzzling that most TAD boundaries in mammalian genomes contain clustered CBS elements with complex orientations of both convergence and divergence.

TAD boundaries emerged as insulator elements that restrain inter-TAD chromatin interactions. Their dysfunction leads to abnormal development and progressive oncogenesis[10–12]. Mammalian clustered CTCF TAD boundaries are more evolutionarily conserved than other genomic regions[13–15]. High insulation score of TAD boundaries is associated with strong CTCF enrichment, and clustered enhancers,

[1]Center for Comparative Biomedicine, Ministry of Education Key Laboratory of Systems Biomedicine, State Key Laboratory of Systems Medicine for Cancer, Joint International Research Laboratory of Metabolic and Developmental Sciences, School of Life Sciences and Biotechnology, Institute of Systems Biomedicine, Shanghai Jiao Tong University, Shanghai 200240, China. [2]WLA Laboratories, Shanghai 201203, China. [3]Laboratory Animal Center, Instrumental Analysis Center, Shanghai Jiao Tong University, Shanghai 200240, China. [4]These authors contributed equally: Xiao Ge, Haiyan Huang. ✉e-mail: qiangwu@sjtu.edu.cn

known as super-enhancers, tend to be insulated by strong boundaries[16,17]. Divergent CBS elements are enriched at TAD boundaries; however, a divergent orientation signature is not strictly required for effective insulation. For example, the characteristics of specific CBS elements can outweigh CBS number and orientation[18]. Finally, insulation potency depends on the local context of CBS elements and robust enhancer-promoter communications can bypass CTCF insulation regardless of the intervening CBS strength[19–22].

As a paradigm for investigating mechanisms of 3D genome folding, the mouse clustered protocadherins (*cPcdh*) comprise three sequential gene clusters, *Pcdh α, β*, and *γ*, encoding 58 isoforms and spanning ~1 million bps (Fig. 1a)[23]. The *Pcdh α* and *γ* clusters consist of 14 and 22 variable first exons, respectively, each of which is *cis*-spliced to a single set of cluster-specific constant exons; whereas the *Pcdhβ* cluster consists of 22 single variable exons (*β1-β22*) with no constant exon. The *Pcdh* promoter choice is determined by CTCF/cohesin-mediated long-range enhancer-promoter looping interactions anchored by convergent or opposite CBS elements[9,24,25]. Their combinatorial expression patterns in single neurons generate enormous cell-surface molecular diversity for assembling complex networks of neuronal connectivity in the brain[26,27]. In addition, precise expression patterns of members of the *HoxD* gene cluster are central for limb development[28,29]. Here, using *Pcdh* and *HOXD* as model genes, we find that outward-oriented CBS elements within clustered CTCF boundaries are crucial for intra-TAD promoter-enhancer interactions and gene regulation. Our data shed interesting light on the organizational principles of the clustered CTCF TAD boundaries in 3D genome architecture and gene regulation.

## Results

### The downstream boundary of *cPcdh* superTAD comprises a cluster of CTCF sites with mixed orientations

During mouse neocortical development, the 58 *cPcdh* genes are organized into a large megabase-sized superTAD comprised of the *Pcdhα* and *Pcdhβγ* TADs (Fig. 1a; Supplementary Fig. 1a–c)[9,30]. At the telomeric boundary of the *cPcdh* superTAD, there is a complex array of eight CTCF sites or CBS elements (*CBS a-h*, Fig. 1a–c). Among these eight CBS elements, *CBSa-e*, *CBSg*, and *CBSh* are in the reverse orientation (Supplementary Fig. 1d). By contrast, a single *CBSf* is in the forward orientation (Supplementary Fig. 1e). We performed chromatin immunoprecipitation followed by high-throughput sequencing (ChIP-seq) experiments with a specific antibody against CTCF using micro-dissected mouse neocortical tissues and found that CTCF is enriched at these elements (Fig. 1c). Interestingly, the *CBS b* and *c* elements form a single peak and *CBS f* and *g* elements form a single peak (Fig. 1c). To see whether these elements are anchors of cohesin loop extrusion, we performed ChIP-seq experiments with a specific antibody against Rad21, a subunit of cohesin, and found that cohesin is co-localized with CTCF at all of these CBS elements, suggesting that cohesin loop extrusion from this boundary may regulate *cPcdh* gene expression (Fig. 1b, c).

We first comprehensively mapped expression patterns of each member of the three *Pcdh* gene clusters at a series of 12 developmental stages by collecting embryos every other day and new pubs every other 2 days. RNA-seq of microdissected neocortical tissues showed that each member of the three *Pcdh* gene clusters is expressed dynamically and that the *cPcdh* genes are most abundantly expressed at the newborn stage (Supplementary Fig. 2). Thus, we used the newborn mouse neocortical tissues in the following gene regulation studies.

Within the *Pcdhα* TAD, members of the *Pcdhα* cluster are activated by a super-enhancer comprised of *HS5-1* and *HS7* (Fig. 1b and Supplementary Fig. 3a, b)[24,31,32]. However, how members of the *Pcdhβγ* clusters are regulated within the *Pcdhβγ* TAD is not clear. To this end, we profiled the chromatin regulatory landscape by assaying for transposase-accessible chromatin with high-throughput sequencing

(ATAC-seq) and identified six highly accessible regions at the telomeric boundary of the *Pcdhβγ* TAD, corresponding to the previously identified DNaseI hypersensitive sites of *HS7L*, *HS5-1aL*, *HS5-1bL*, *HS18*, *HS19-20*, and *HS21* (Fig. 1c and Supplementary Fig. 3c, d)[24,33]. Finally, we performed H3K27ac and H3K4me1 ChIP-seq and found, except the last one (*HS21*), all of the other five clustered *HS* elements are marked by H3K27ac and H3K4me1, suggesting that they form a super-enhancer for members of the *Pcdhβγ* gene clusters (Fig. 1c).

### The *Pcdhβγ* TAD boundary comprises a super-enhancer in mice in vivo

We dissected the putative enhancers in mice in vivo by CRISPR/Cas9-based DNA-fragment editing system (Supplementary Fig. 4a–c)[9] to test their functionality. We first deleted the *HS7L* element (Supplementary Fig. 4b, c) and found, surprisingly, it does not lead to decreased levels of *cPcdh* gene expression in neocortical and olfactory tissues (Fig. 1d, Supplementary Fig. 4e–g, and Supplementary Fig. 5a). Because *HS5-1aL* is in the *Pcdhγ* constant coding region and its deletion will destroy the transcript integration, we mutated the *CBSa* element within *HS5-1aL* by homologous recombination using CRISPR with an ssODN donor. We found that *CBSa* mutation or deletion abolished CTCF enrichments (Supplementary Fig. 4d). Interestingly, it only results in a moderate decrease of expression levels of members of the *Pcdhβγ*, but not *α*, clusters (Fig. 1e, Supplementary Figs. 4e, 5b), suggesting a limited role of *CBSa* in tethering *HS5-1aL* to *Pcdhβγ*.

We next deleted the entire region covering all of the other three putative enhancers (from *HS5-1bL* to *HS19-20*, *CBSb-e*) in mice and found that, in contrast to *HS7* and *HS5-1aL*, this large deletion leads to a significant decrease of levels of *Pcdhβγ* gene expression, suggesting that enhancers for members of *Pcdhβγ* clusters reside in this region (Fig. 1f and Supplementary Fig. 5c). We then dissected this region in great details and found that deletion of the large DNA fragment covering both *HS5-1bL* and *HS18* (*CBSb-d*) results in a significant decrease of levels of *Pcdhβγ* gene expression (Fig. 1g and Supplementary Fig. 5d). In addition, deletion of the DNA fragment covering both *HS18* and *HS19-20* (*CBSde*) also results in a significant decrease of levels of *Pcdhβγ* gene expression (Fig. 1h and Supplementary Fig. 5e).

Finally, we deleted each of these three elements individually in mice and found that, surprisingly, deletion of *HS5-1bL* (*CBSbc*) has no effect on *Pcdhβγ* gene expression (Fig. 1i and Supplementary Fig. 5f). By contrast, deletion of either *HS18* (*CBSd*) or *HS19-20* (*CBSe*) result in a significant decrease of levels of *Pcdhβγ* gene expression (Fig. 1j, k and Supplementary Fig. 5g, h). These data suggest that these two H3K27ac-enriched elements play a leading role in activating *Pcdhβγ* genes in vivo in the mouse brain. Consistently, 4C experiments revealed close contacts between *Pcdhβγ* and *HS18-20* (Supplementary Fig. 6a, b). Unlike *HS5-1*, which is brain-specific, *HS18-20* are enriched with H3K27ac and H3K4me3 marks in multiple tissues (Supplementary Fig. 6c, d), explaining why members of the *Pcdhβγ* gene clusters are expressed in both neural and non-neural tissues. Taken together, these data suggest that the downstream boundary of the *Pcdhβγ* TAD is a super-enhancer comprising a cluster of enhancers among which *HS18* and *HS19-20* are the most important.

### *HS5-1bL* is a *Pcdhγc3*-specific insulator in mice in vivo

*Pcdhγc3* is the most abundantly expressed isoform in the mouse neocortex and has a unique role in dendrite arborization (Supplementary Fig. 2e)[34]. However, the underlying mechanism remains unclear. To this end, we performed QHR-4C experiments and found that *Pcdhγc3* is in close contacts with *HS5-1bL*, *HS18*, and *HS19–20* (Fig. 2a). Consistently, deletion of *HS18* and/or *HS19-20* results in a significant decrease of levels of *Pcdhγc3* gene expression, suggesting that *HS18* and *HS19-20* are enhancers for *Pcdhγc3* (Fig. 2b–d). Surprisingly, deletion of *HS5-1bL* in mice in vivo leads to a significant increase of levels of *Pcdhγc3*, but not *γc4* or *γc5*, gene expression

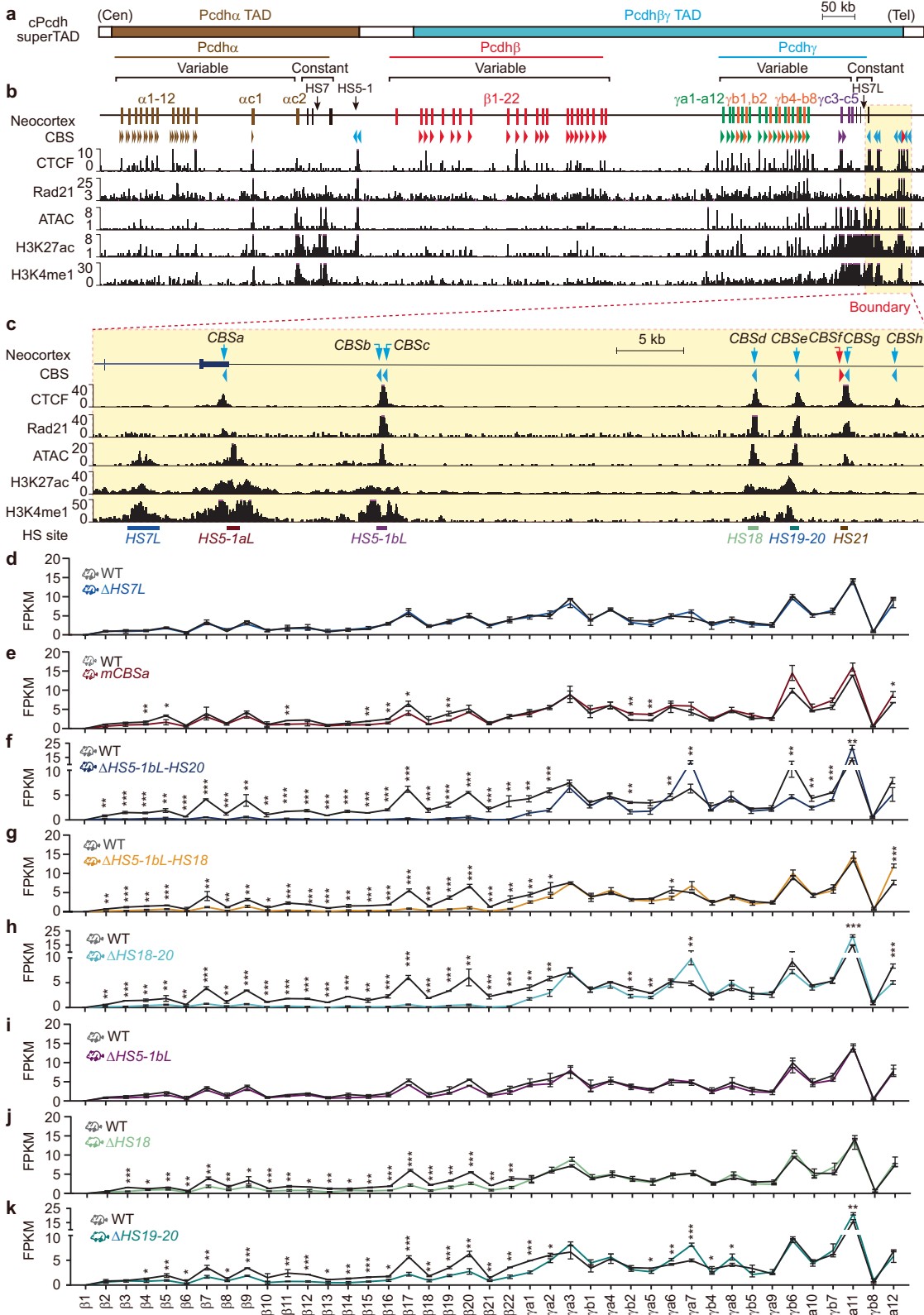

(Fig. 2e), suggesting that *HS5-1bL* is a putative insulator specifically blocking the activation of *Pcdhγc3* by the *HS18* and *HS19-20* enhancers. To this end, we performed 4 C experiments with *Pcdhγc3* as an anchor using neocortical tissues of the *HS5-1bL* deleted mice and found a significant increase of long-distance chromatin interactions between *Pcdhγc3* and *HS18–20* (Fig. 2f). To confirm this aberrantly increased

chromatin contacts upon deletion of *HS5-1bL*, we performed 4 C with *HS18* or *HS19–20* as an anchor and indeed observed increased chromatin contacts with *Pcdhγc3* (Fig. 2g). In conjunction with no expression increase of all other members of the *Pcdhβγ* gene clusters upon deletion of *HS5-1bL* (Fig. 1i), we conclude that *HS5-1bL* is a *γc3*-specific insulator.

**Fig. 1 | The downstream *cPcdh* superTAD boundary comprises clustered enhancers for *Pcdhβγ* gene regulation. a** Schematics of the mouse *cPcdh* locus. The mouse *Pcdh α, β*, and *γ* gene clusters are closely-linked within the *cPcdh* superTAD, which is divided into two TADs of *Pcdhα* and *Pcdhβγ*. *Pcdh α* and *γ* gene clusters contain 14 and 22 variable first exons, respectively, each of which is alternatively spliced to a single set of three downstream constant exons. The *Pcdhβ* gene cluster contains 22 variable exons but with no constant exon. Two *Pcdhα* enhancers, *HS5-1* and *HS7*, are indicated by vertical black arrows. **b** ChIP-seq of CTCF, Rad21, H3K27ac, and H3K4me1 as well as ATAC-seq profiles of the *cPcdh* locus in the mouse neocortex. Arrowheads indicate CBS elements with orientations. Within the *Pcdhα* TAD, each *Pcdhα* alternate promoter is flanked by two forward CBS elements, the downstream enhancer of *HS5-1* is flanked by two reverse CBS elements. Within the *Pcdhβγ* TAD, each *Pcdh β* or *γ* promoter is associated with one forward CBS, except for *β1, γc4*, and *γc5* which have no CBS, as well as *γc3* which has two

forward CBS elements. **c** Close-up of the *cPcdh* superTAD downstream boundary marked by a red dotted rectangle in (**a, b**). This boundary comprises eight CBS elements, of which only *CBSf* is in the reverse orientation, and six ATAC-seq peaks, of which *HS7L, HS5-1bL, HS18*, and *HS19-20* are knocked out individually or in combinations in mice. *HS5-1aL* is mutated at the *CBSa* motif (*mCBSa*) due to its location in the coding region. **d**–**k** Expression levels of members of the *Pcdh β* and *γ* gene clusters in the mouse neocortex from Δ*HS7L*, *mCBSa*, Δ*HS5-1bL-HS20*, Δ*HS5-1bL-HS18*, Δ*HS18-20*, Δ*HS5-1bL*, Δ*HS18*, or Δ*HS19-20* homozygous mice compared to their wild-type (WT) littermates. FPKM, fragments per kilobase of exon per million fragments mapped. Data as mean ± SD; two-tailed Student's *t* test. *$p < 0.05$, **$p < 0.01$, ***$p < 0.001$. For each mouse line, $n = 4$ biologically independent samples; for their WT littermate controls, $n = 2$. Source data are provided as a Source Data file.

## Clustered *Pcdhs* are expressed from H3K9me3-enriched facultative heterochromatins of superTAD

We next mapped the chromatin landscape of a wide variety of histone modifications in the *cPcdh* superTAD and found that this region is enriched for both the repressive mark of H3K9me3 and the active mark of H3K36me3 (Fig. 3a), suggesting that *cPcdhs* are expressed from facultative heterochromatin region of the superTAD. In addition, expression of each member of the three *Pcdh* gene clusters is strongly correlated with active marks of H3K4me3 and H3K9ac (Fig. 3b and Supplementary Figs. 7, 8). Moreover, this region is also marked by the repressive histone marks of H3K9me2 in the mouse neocortex (Fig. 3c and Supplementary Fig. 9). Specifically, the coding region of each member of the *Pcdh* clusters is correlated with a peak of H3K9me2 (Supplementary Fig. 9) but not H3K9me3 (Supplementary Fig. 10). To further investigate mechanisms of *cPcdh* gene regulation, we analyzed methylated DNA immunoprecipitation sequencing (MeDIP-seq) data and found that each member of the H3K9me3-enriched *cPcdhs* is hypermethylated on the body of each variable exon (Fig. 3c, Supplementary Fig. 11a–d). We noted that *Pcdhβ1* differs from other *cPcdh* members in that its DNA-methylation peak is much closer to the transcription start site (Supplementary Fig. 11e), explaining why it is silenced in the mouse neocortex (Fig. 3b).

## Outward-oriented *CBSf* is the key boundary CTCF site

All of the *HS5-1aL, HS5-1bL, HS18*, and *HS19-20* enhancers contain reverse-oriented CBS elements (*CBSa-e*, Fig. 1c) that are enriched for cohesin and are in close contacts with the forward-oriented CBS elements of the *Pcdhβγ* promoters via CTCF/cohesin-mediated chromatin loops. In addition, there are two more reverse-oriented CBS elements, *CBSg* and *CBSh*, located immediately downstream of this super-enhancer. However, there is a single forward-oriented *CBSf* element located between the super-enhancer and the two reverse-oriented *CBS g* and *h* elements (Fig. 4a). To this end, we generated a *CBSf* deletion mouse line (Fig. 4a and Supplementary Fig. 12a) and confirmed the abolishment of CTCF and cohesin enrichments at the location of *CBSf* but not *CBSg* (Fig. 4a, b). In addition, CTCF and cohesin binding at other CTCF sites within this boundary is not altered by *CBSf* deletion (Supplementary Fig. 12b).

Remarkably, deletion of the forward-oriented *CBSf* element results in a significant decrease of expression levels of members of the *Pcdhβγ* gene clusters (Fig. 4c and Supplementary Fig. 13a), demonstrating that the single forward-oriented *CBSf* plays a crucial role in the regulation of *Pcdhβγ* gene expression. To investigate the underlying mechanism, we performed a series of QHR-4C experiments with members of the *Pcdhβγ* gene clusters as anchors and found a significant decrease of long-distance chromatin contacts between the super-enhancer and members of the *Pcdhβγ* gene clusters (Fig. 4d). In particular, we observed a significant increase of long-distance chromatin contacts with *CBSh* with a sharp edge at the location of *CBSf* (Fig. 4e). To confirm this, we performed reciprocal capture

experiments with *CBSh* as an anchor and observed a significant increase of long-distance chromatin contacts with members of the *Pcdhβγ* gene clusters (Supplementary Fig. 12c). In addition, we profiled the histone modifications and found decrease in enrichments of repressive marks, H3K9me3 and H3K9me2, but not of active marks, H3K36me3, H3K4me3, and H3K9ac (Supplementary Figs. 14–17).

Finally, combined deletion of both *CBSf* and *CBSg* elements (Supplementary Fig. 12d) results in a similar phenotype (Supplementary Figs. 18, 12e, 13b). As a control, single deletion of *CBSg* results in a less decrease in *Pcdhβγ* expression (Supplementary Figs. 12f–h and 19). Taken together, we conclude that a single outward-oriented *CBSf* element within the clustered CTCF TAD boundary plays a key role in *Pcdhβγ* gene regulation and long-distance intra-TAD chromatin interactions.

## Outward-oriented CBS elements within a clustered CTCF TAD boundary are crucial for *HOXD13* expression

To see whether it is a general phenomenon for a key role of the outward-oriented boundary CBS element in intra-TAD chromatin interactions and gene regulation, we investigated the *HOXD* left boundary, which has a complex array of six conserved clustered CBS elements (*CBS1-6*, Supplementary Fig. 20a–d), of the human centromeric TAD (C-DOM) (Fig. 5a, b)[28,29]. The human *HOXD* locus is located at the boundary between C-DOM and T-DOM (telomeric TAD) (Supplementary Fig. 21a, b). It contains nine *HOXD* genes, of which only *HOXD13* is associated with a cluster of reverse-oriented CBS elements (*CBS8-12*, Fig. 5c). By contrast, *HOXD8* and *HOXD9* have two forward-oriented CBS elements (*CBS13,14*) and *HOXD10-12* and *HOXD1-4* have no CBS element (Fig. 5a–c and Supplementary Fig. 20a). We analyzed Hi-C and found that there are strong long-distance chromatin interactions between the human *HOXD13* and the left boundary of C-DOM, consistent with the convergent rule of forward-reverse CBS elements (Supplementary Fig. 21a, b). Interestingly, within the left boundary, there are two outward-oriented CBS elements (*CBS3* and *CBS5*, Fig. 5b and Supplementary Fig. 21b).

We screened single-cell clones for deletion of both *CBS3* and *CBS5* elements and obtained three homozygous double-knockout cell clones (Δ*CBS3 + 5*, Supplementary Fig. 20g). We first confirmed the loss of CTCF enrichments at both *CBS3* and *CBS5* (Fig. 5a, b). We then performed 4C with *HOXD13* as an anchor and found that there is a significant increase of chromatin interactions with the left boundary upon deletions of the two outward-oriented CBS elements (Fig. 5d). These increased chromatin interactions were confirmed by reciprocal 4C experiments with *CBS2* or *CBS4* as an anchor (Fig. 5e, f). By contrast, there is a significant decrease of chromatin interactions with proximal sequences (Fig. 5d), which is confirmed by reciprocal 4C experiments with *GT2* or *LNPK* as an anchor (Supplementary Fig. 21c, d). In addition, we performed RNA-seq experiments and found that deletion of both outward-oriented CBS elements results in a significant decrease of expression levels of *HOXD13* but not *HOXD12-1*, indicating that the two

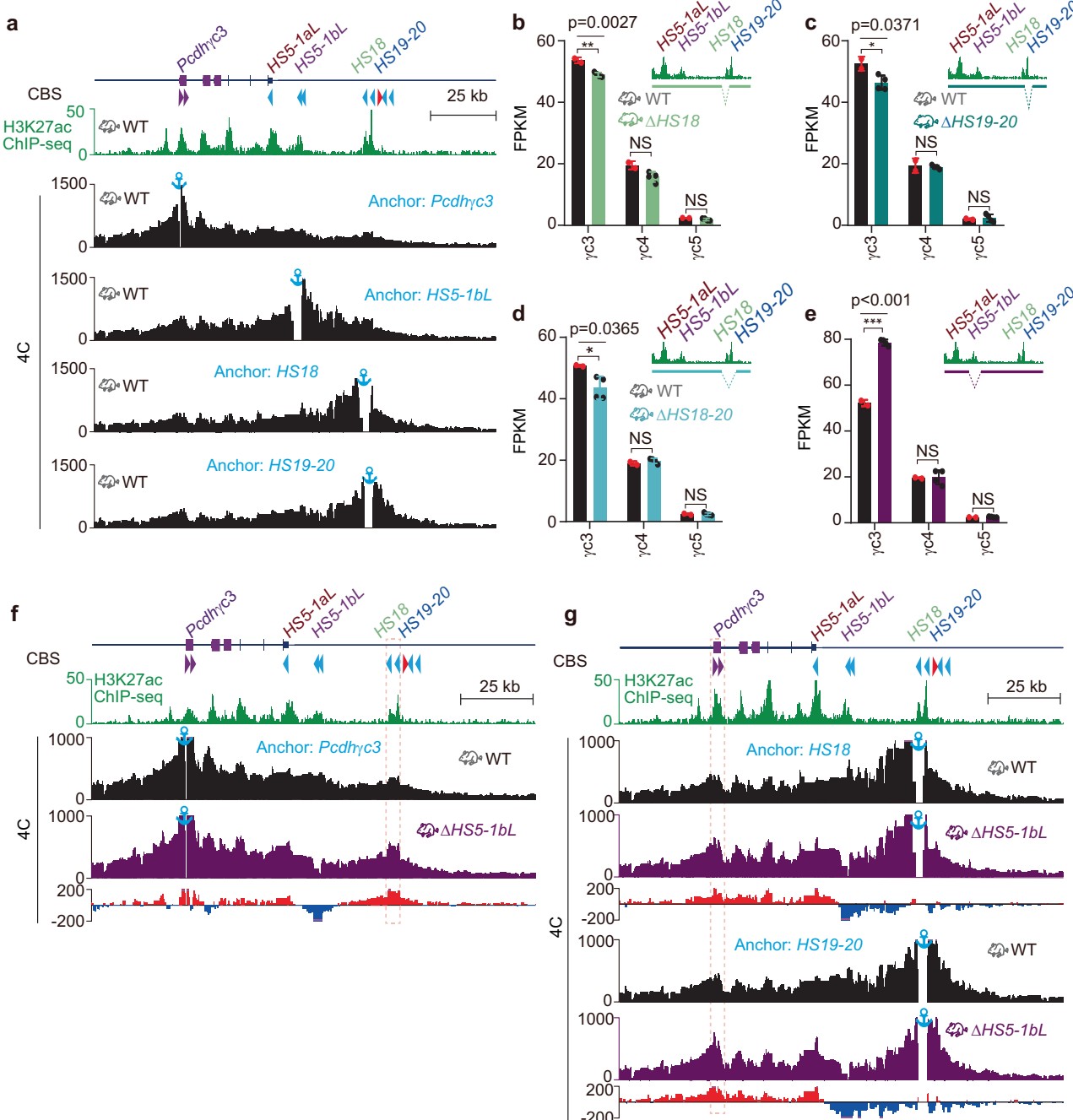

**Fig. 2 | HS5-1bL is an insulator specific for Pcdhyc3. a** 4C profiles using a repertoire of elements as anchors showing close contacts between *Pcdhyc3* and *HS5-1bL*, *HS18*, or *HS19-20* in wild-type (WT) mice. **b–e** Expression levels of the *Pcdhy* c-type genes in the mouse neocortex from *ΔHS18* (**b**), *ΔHS19-20* (**c**), *ΔHS18-20* (**d**), *or ΔHS5-1bL* (**e**) homozygous mice compared to their wild-type littermates. Note the significant increase of expression levels of *Pcdhyc3* upon deletion of *HS5-1bL*. Data as mean ± SD; two-tailed Student's *t* test. *\*p* < 0.05, *\*\*p* < 0.01, *\*\*\*p* < 0.001. For each mouse line, *n* = 4 biologically independent samples; for their WT littermate controls, *n* = 2. Source data are provided as a Source Data file. **f** 4C profiles using the *Pcdhyc3* promoter as an anchor, showing increased chromatin interactions between *Pcdhyc3* and its *HS18-20* enhancers in *ΔHS5-1bL* homozygous mice compared to their WT littermates. Differences (*ΔHS5-1bL* versus WT) are shown under the 4 C profiles. **g** 4C profiles using *HS18 or HS19-20* as an anchor, showing increased chromatin interactions between *Pcdhyc3* and *HS18-20* in *ΔHS5-1bL* mice compared to their WT littermates. Differences (*ΔHS5-1bL* versus WT) are shown under the 4C profiles.

outward-oriented CBS elements play a key role in *HOXD13* gene regulation (Fig. 5g). However, different from the *Pcdhβy* genes which are associated with CBS elements, there is no CBS element associated with the LNPK gene within C-DOM and its expression is not altered upon deletion of outward-oriented CBS elements (Fig. 5g). Finally, single deletions of the *CBS3* or *CBS5* element demonstrated a cooperation role between the two outward-oriented CBS elements in *HOXD13* gene regulation (Supplementary Fig. 22).

## Discussion

The enormous connectivity of billions of neurons is intricately related to the precise cell-specific expression patterns of *cPcdh* genes in the brain. Specifically, the isoform-promiscuous *cis* hetero-dimerization and isoform-specific *trans* homo-dimerization of combinatorially expressed cPcdhs mediate cell-specific recognition between neuronal membranes, resulting in ingenious dendrite self-avoidance and axonal coexistence[27,35–42]. Intriguingly, chromatin

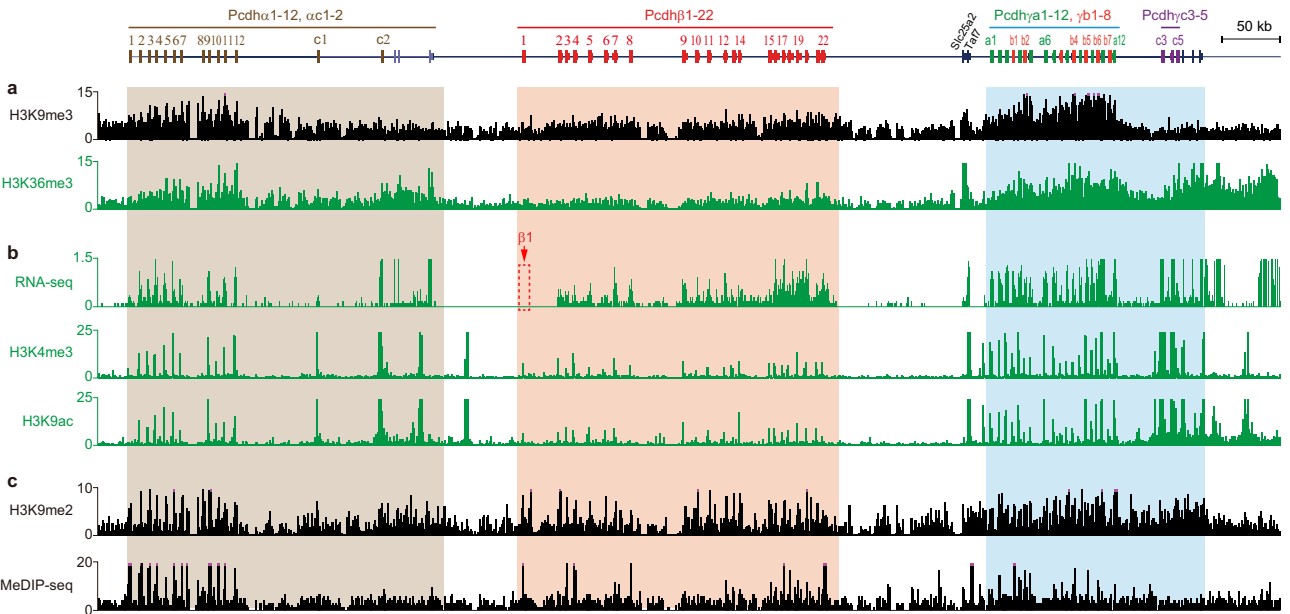

**Fig. 3 | The *cPcdh* genes are expressed from H3K9me3/H3K36me3-enriched facultative heterochromatin domains. a** H3K9me3 and H3K36me3 ChIP-seq profiles of the three *Pcdh* clusters in mouse neocortex, showing colocalization of both the active mark of H3K36me3 and inactive mark of H3K9me3. **b** RNA-seq as well as H3K4me3 and H3K9ac ChIP-seq profiles of the *cPcdh* clusters in mouse neocortex, showing a strong correlation between each *cPcdh* gene and the enrichment of active marks. **c** The H3K9me2 ChIP-seq and MeDIP-seq profiles of the *cPcdh* clusters, showing enrichments of the repressive mark of H3K9me2 and DNA methylation at most *cPcdh* genes in the mouse neocortex.

establishes a stable *cPCDH* expression pattern for single neurons in the brain during human fetal development[43]. In addition, the proper spatiotemporal expression of the *HoxD* genes is central for limb development[29,44]. Here, we first comprehensively mapped *cPcdh* expression patterns in the neocortex during mouse brain development. We then dissected, through sequential and combinatorial deletion of candidate enhancer elements, how *cPcdhs* are regulated by a downstream clustered CTCF boundary super-enhancer. In addition, we further deleted outward-oriented CBS elements within *Pcdh* and *HOXD* clustered CTCF TAD boundaries, which are associated with dozens of human pathogenic copy number variations (CNVs) (Supplementary Data 1). In conjunction with chromosome conformation and gene expression analyses, we found that outward-oriented CBS elements within clustered CTCF TAD boundaries are crucial for intra-TAD promoter-enhancer interactions and proper gene regulation (Fig. 5h). Unlike inward-oriented CBS elements, which directly make intra-TAD spatial contacts, such as between enhancers and promoters, the outward-oriented CBS elements may block extruding cohesin sliding from the outside of the TAD to prevent aberrant inter-TAD chromatin interactions (Fig. 5h).

The characteristics of the clustered CTCF TAD boundaries and their exact nature of insulation activities are under intense investigation[13,18,20,22,29,44–48]. The CBS elements within a clustered CTCF TAD boundary between *Pax3* and its distal enhancer cooperate redundantly for robust insulation[18]. In addition, various combinations of clustered CBS elements in distinct orientations could function as topological insulators when inserted between distal enhancers and their target promoters[20,22,49]. Moreover, the CBS elements within a clustered CTCF TAD boundary located between the C-DOM and T-DOM (telomeric domain) of *HoxD* have various functions for controlling its spatiotemporal expression patterns during limb development[29,44]. Here, we show that the outward-oriented CBS elements of the left boundary of C-DOM are crucial for maintaining intra-TAD chromatin interactions and *HOXD13* gene expression (Fig. 5). Finally, consistent with the crucial role of outward-oriented CBS

elements, an inversion leading to increased numbers of outward-oriented CBS elements within a clustered CTCF TAD boundary results in enhanced insulation activity and altered gene expression[50].

CTCF plays a key role in the formation of insulation at TAD boundaries, where in many cases multiple CBS elements are arranged in a clustered array with complex and mixed orientations, such as the telomeric downstream boundary of the *Pcdhβγ* TAD and the centromeric upstream boundary of the C-DOM of the *HOXD* cluster. According to the convergent rule, the inward-oriented CBS elements within the clustered CTCF TAD boundaries are essential for intra-TAD chromatin interactions[8,9] (Fig. 5h). Thus, chromatin contacts such as long-distance enhancer-promoter interactions primarily occur within the TADs[51,52]. However, recent studies provide increasing evidence for inter-TAD communications between neighboring TADs[50]. Although TADs are likely a statistical model from population cells, there is clear evidence of clustering of CBS elements at TAD boundaries[13,15,48]. On one hand, the outward-oriented CBS elements could be crucial for intra-TAD chromatin interactions. On the other hand, the outward-oriented CBS elements function as crucial insulators blocking aberrant chromatin interactions from neighboring TADs as we observed a sharp edge for increased chromatin interactions beyond the outward CBS element (Fig. 4d, e). In the case of clustered CTCF TAD boundary of *Pcdhβγ*, the extruding cohesin complexes from the downstream TAD cannot pass through the forward *CBSf* element. However, upon deletion of the outward *CBSf* element, these upward-extruding cohesin complexes pass through the entire TAD boundary and eventually anchored at variable promoters, leading to an increase of chromatin interactions between *Pcdhβγ* genes and *CBSh* region.

The structure of TADs and loops is closely related to gene expression. Their formation is thought to be resulted from CTCF-anchored active cohesin "loop extrusion". Specifically, cohesin-extruded loops anchored between convergent forward-reverse CBS elements can enhance or stabilize enhancer-promoter spatial contacts, which may or may not activate gene expression[53–56]. In addition, both the spatial and linear distances between distal enhancers and target promoters could be crucial for gene regulation[57–59]. Our study

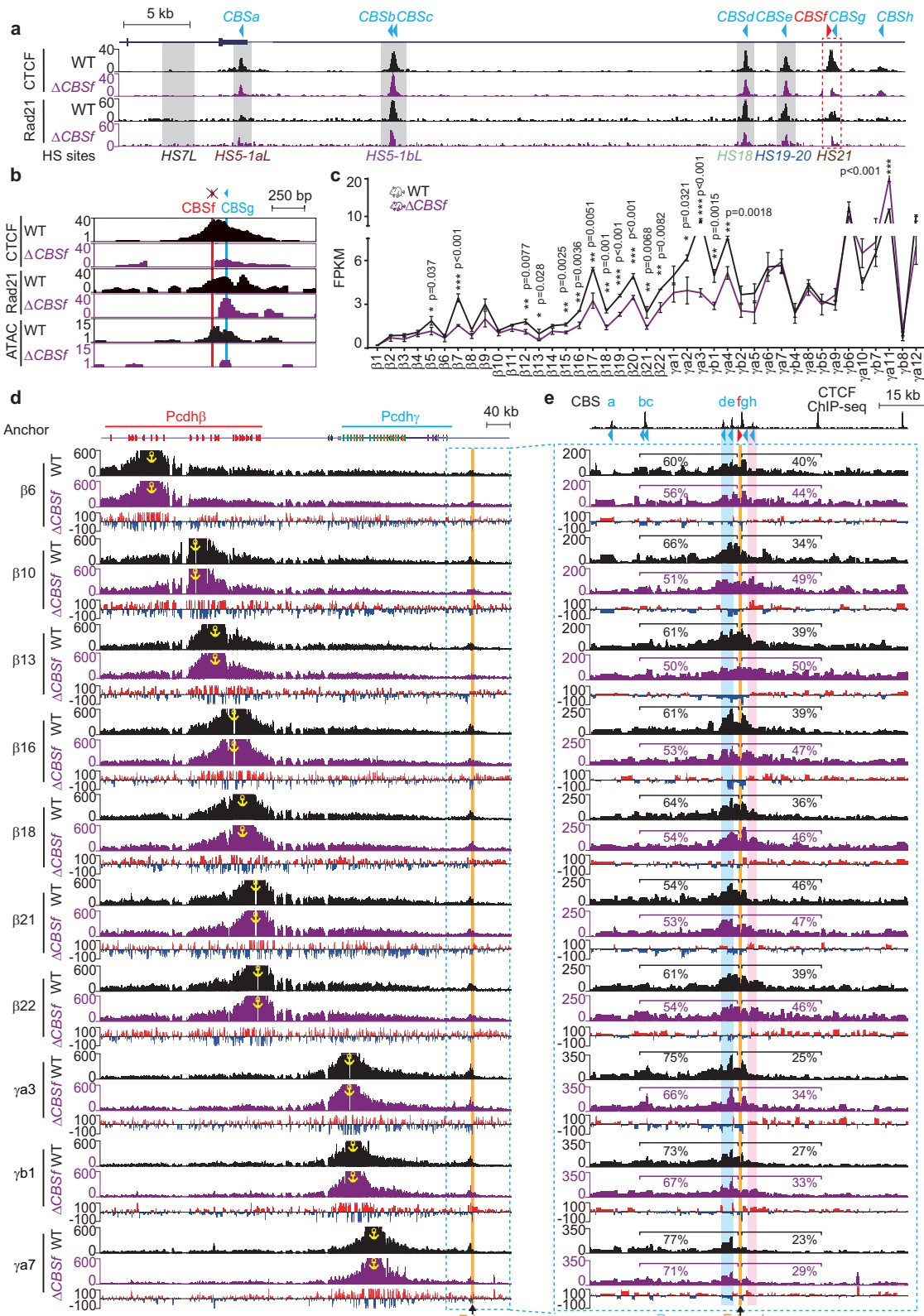

**Fig. 4 | Outward-oriented *CBSf* regulates *Pcdhβγ* by blocking their aberrant looping from the outside of the TAD boundary. a** CTCF and Rad21 ChIP-seq profiles of the *Pcdhβγ* downstream TAD boundary in *CBSf*-deleted (Δ*CBSf*) homozygous mice compared to their wild-type (WT) littermates. **b** Close-up of CTCF and Rad21 ChIP-seq as well as ATAC-seq profiles in Δ*CBSf* mice compared to their WT littermates. **c** RNA-seq showing decreased expression levels of *Pcdhβγ* upon *CBSf* deletion. Data as mean ± SD, *\*p* < 0.05, *\*\*p* < 0.01, *\*\*\*p* < 0.001; two-tailed Student's *t*

test. For *CBSf* deletion mouse line, *n* = 4 biologically independent samples; for their WT littermate controls, *n* = 2. Source data are provided as a Source Data file. **d, e** 4C profiles using a repertoire of *Pcdhβγ* promoters as anchors, showing decreased chromatin interactions with *HS18-20* enhancers (highlighted in blue, (**e**)) and increased chromatin interactions beyond *CBSf* (highlighted in pink, (**e**)). Note a sharp edge for the transition from decrease to increase at the location of *CBSf*. Differences (Δ*CBSf* versus WT) are shown under the 4C profiles.

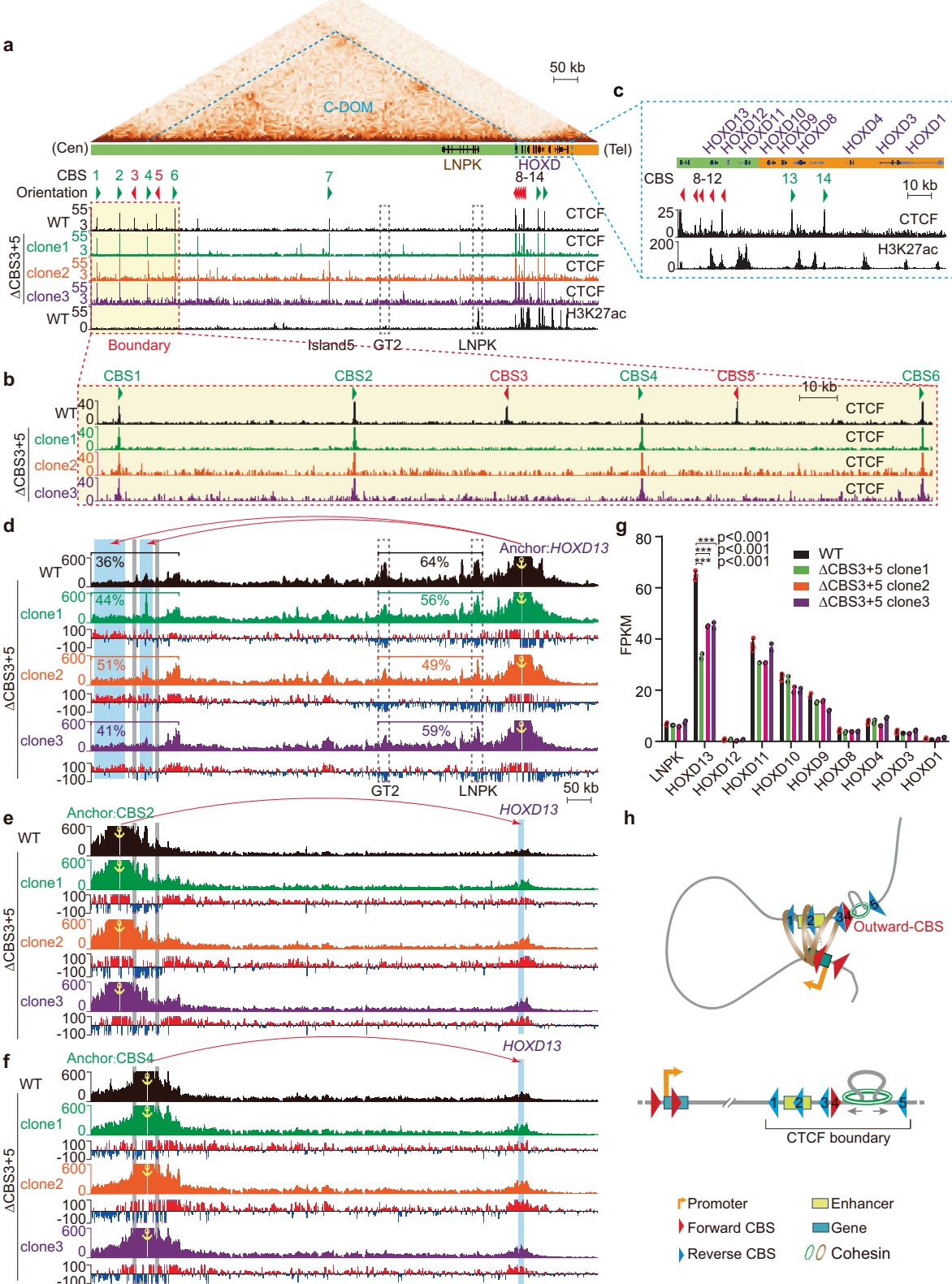

on *Pcdhγc3* regulation is consistent with this idea. *Pcdhγc3*, the most abundantly expressed *cPcdh* isoform in the mouse neocortex (Supplementary Fig. 2e), plays a crucial role in dendrite arborization[34]. It is puzzling why *HS5-1bL* deletion leads to a significant increase of expression levels of only *γc3*, but not other members, of *cPcdh* genes, suggesting that it is a *γc3*-specific insulator (Figs. 1i, 2e and Supplementary Fig. 4e). It has been shown that gene activation is not only

dependent on promoter-associated CBS elements, but also on the density of enhancers of the genomic context[32,57]. Consistently, only *γc3*, but not *γc4* and *γc5*, is associated with two forward-oriented CBS elements (Fig. 2a) despite the fact that, compared with other members of *Pcdhβγ* clusters, all of the three *Pcdhγc* genes are located close to the downstream distal enhancers of *HS18-20*, which contain two reverse-oriented CBS elements (Fig. 1b). A "double clamp"

**Fig. 5 | Double knockout of outward-oriented *CBS3* and *CBS5* elements down-regulates *HOXD13* gene expression. a** CTCF and H3K27ac ChIP-seq profiles of the *HOXD* centromeric regulatory TAD (C-DOM) in wild-type (WT) and in *CBS3* and *CBS5* double knockout (Δ*CBS3 + 5*) homozygous single-cell clones. The left boundary (red-dotted rectangle) comprises a cluster of CTCF sites, *CBS1-6*, of which *CBS3* and *CBS5* are in the outward orientation. Arrowheads indicate CBS elements with orientations. **b** Close-up of CTCF ChIP-seq profiles at the C-DOM left boundary marked by red dotted rectangle in (**a**). Note the abolishment of CTCF peaks at the locations of *CBS3* and *CBS5* in the three single-cell deletion clones. **c** Close-up of CTCF and H3K27ac ChIP-seq profiles at the *HOXD* cluster in (**a**). Note that the five reverse-oriented CBS elements are associated with *HOXD13* but not with other *HOXD* genes. **d** 4C profiles using *HOXD13* as an anchor in Δ*CBS3 + 5* single-cell clones compared to WT clones, showing increased chromatin interactions with the left boundary and decreased chromatin interactions with the H3K27ac-enriched *GT2* or *LNPK* element upon double knockout. **e, f** 4C profiles using *CBS2* (**e**) or *CBS4* (**f**) as an anchor confirming increased chromatin interactions with *HOXD13*. **g** Expression levels of *HOXD* genes in three Δ*CBS3 + 5* single-cell clones compared to WT clones. Data as mean ± SD, *$p < 0.05$, **$p < 0.01$, ***$p < 0.001$; two-tailed Student's *t* test. For each deletion clone, $n = 2$ biologically independent samples; for their WT controls, $n = 4$. Source data are provided as a Source Data file. **h** A boundary model illustrating the role of outward-oriented CTCF sites (CBS elements) in intra-TAD chromatin contacts and gene regulation. TAD boundary normally contains clustered CTCF sites with complex orientations of both divergence and convergence. The outward-oriented CTCF sites function as an important barrier for outside cohesin to ensure proper and balanced intra-TAD chromatin interactions and gene expression.

interaction mechanism of the two forward-reverse CBS pairs, similar to the activation of members of the *Pcdhα* gene cluster[24], may be utilized to activate *γc3*. This explains why only *γc3*, but not *γc4* and *γc5*, expression levels are enhanced upon deletion of the *HS5-1bL* (*CBSbc*) insulator.

*HOXD13* showed strong interactions with two H3K27ac-marked regions of *LNPK* and *GT2*. We observed a significant decrease of chromatin interactions with *LNPK* and *GT2* upon removal of outward-oriented CBS elements (Fig. 5d), explaining the decreased levels of *HOXD13* gene expression. This suggests that the outward-oriented CBS boundary elements of the C-DOM may ensure proper interactions with *LNPK* and *GT2* for precise control of *HOXD13* expression patterns. Our data highlight a significant role of the outward-oriented CBS elements in providing optimal insulation for restraining aberrant inter-TAD interactions and maintain proper intra-TAD promoter-enhancer interactions in the *Pcdh* and *HoxD* clusters. We analyzed genome-wide CBS patterns at TAD boundaries and found that 85.7% have outward-oriented CBS elements (Supplementary Data 2, 3). However, whether it could be generalized to the entire genome awaits further studies.

We observed that H3K9me3 marks correlate with the monoallelic *cPcdh* gene expression and that H3K9me2 marks correlate with both monoallelic and biallelic *cPcdh* gene expression[26]. The correlation between H3K9me3 enrichments and DNA methylation levels in both monoallelic and biallelic *cPcdh* genes may be coordinated by UHRF1[60]. Interestingly, almost every member of the *cPcdh* genes is marked by both repressive (H3K9me3 and H3K9me2) and active (H3K36me3) marks (Fig. 3), suggestive of a facultative heterochromatic state related to monoallelic *cPcdh* gene expression. Whether the overlap of H3K9me3 and H3K36me3 is resulted from allelic differences of the epigenome and/or admixtures of distinct subpopulation cells awaits further exploration.

## Methods
### Cell culture
Human HEK293T cells were cultured in DMEM (Hyclone) supplemented with 10% (v/v) FBS (Gibco) and 10 U/ml penicillin-streptomycin (Gibco, 15140122). Human HEC-1-B cells were cultured in MEM (Hyclone) supplemented with 200 mM L-glutamine, Earle's balanced salt solution (Hyclone, SH30024.01), 1 mM sodium pyruvate (Sigma, SLBP4879V), 10% fetal bovine serum, and 10 U/ml penicillin-streptomycin. All cells were incubated at 37 °C and 5% $CO_2$ in a humidified incubator, and passaged every 3 days.

### CRISPR screening of CBS-deletion single-cell clones
*CBS3* and *CBS5* within the clustered CTCF TAD boundary of the *HOXD* locus were sequentially knocked out by two steps of CRISPR/Cas9-mediated DNA fragment editing (Supplementary Data 4)[9,61]. In the first step, *CBS3* or *CBS5* was deleted in HEK293T cells to generate homozygous *CBS3* or *CBS5* knockout cell clones. In the second step, a *CBS3* knockout cell clone was used to delete *CBS5* and a *CBS3* knockout cell clone was used to delete *CBS3*, to generate cell clones homozygous for *CBS3* and *CBS5* double knockout (ΔCBS3 + 5).

For each step of CRISPR editing, cells were grown to ~80% confluence in 6-well tissue culture plates and transfected with 1.5 μg of pcDNA3.1-Cas9 and 1.5 μg dual sgRNA expression plasmids per well by Lipofectamine 3000 (Invitrogen, L3000015). Two days after transfection, 2 μg/ml puromycin was added to the cell growth medium and maintained for 5 days to select the transfected cells. The transfected cells were suspended into single-cell solutions, diluted and plated into 96 well plates. Two weeks later, single-cell clones were picked and genotyped by PCR with specific primers (Supplementary Data 5). Positive clones were Sanger sequenced.

The sgRNA expression plasmids were constructed as previously described[9,61] with paired oligonucleotides (Supplementary Data 5). For each sgRNA, a forward oligonucleotide with the (ACCG-5′) overhang and a reverse complementary oligonucleotide with the (AAAC-5′) overhang were annealed and cloned into the Bsa I site of pGL3-U6-sgRNA-PGK-Puro vector for sgRNA transcription[62]. All sgRNA-expressed plasmids were confirmed by Sanger sequencing.

### In vitro synthesis of Cas9 mRNA and sgRNAs for micro-injection
A T7 promoter-driven Cas9 expression plasmid[63] was linearized by *Xba* I, purified and in-vitro transcribed using the T7 Ultra Kit (Thermo-Fisher, AM1345). The transcribed Cas9 mRNA was treated with Turbo DNase to remove template DNA and added with poly-A tail. The sgRNA template was generated by PCR amplification with specific primers (Supplementary Data 5) to contain a T7 promoter, a 20 nt target sequence, and a scaffold region[49]. The amplified product was purified and transcribed using the MEGAshortscript T7 Kit (ThermoFisher, AM1354). The transcribed sgRNA was treated with Turbo DNase. Both sgRNA and Cas9 mRNA were purified by the MEGAclear Kit (Thermo-Fisher, AM1908), eluted into elution buffer and stored at −80 °C before usage.

### Generation of CBS-deletion mice
Mice were maintained in an SPF mouse facility at 23 °C, with a humidity between 40 and 60%, with a 12 h (7:00–19:00) light/12 h (19:00–7:00) dark cycle. All the mouse experiments were approved by the Institutional Animal Care and Use Committee (IACUC) of Shanghai Jiao Tong University (Protocol#: 1602029).

C57BL/6J mice were used as embryo donors and ICR mice were used as foster mothers. 6-week-old C57BL/6J female mice were super-ovulated and mated with the sterilized C57BL/6J male mice to produce enough embryos. Zygotes were collected from the oviducts of the super-ovulated C57BL/6J female mice, washed with 200 μg/ml hyaluronidase (Sigma, V900833) and incubated in the M2 medium (Sigma, M7167) in a 5% $CO_2$ incubator at 37 °C for 1 h. Viable embryos were then microinjected with a solution containing 100 ng/μl of Cas9 mRNA and 50 ng/μl each of two sgRNAs targeting each fragment and recovered in a 5% $CO_2$ incubator at 37 °C for 1 h. The injected live embryos were transplanted into the oviducts of the pseudo-pregnant ICR female

mice under a stereoscopic microscope, with each ICR mouse receiving 25–30 embryos. Post-surgery mice were maintained at a 37 °C heating plate for 1 h for recovery before being transferred back for regular housing until the birth of the F0 mice. Mice exhibiting signs of dystocia at day 20 received C-section rescue.

The chimeric F0 mice were screened 10 days later for desired deletions by PCR genotyping with specific primers (Supplementary Data 5) and the chimeric mutant mice were confirmed by Sanger sequencing. The F0 mice of 2-month old with desired deletions were mated with wild-type C57BL/6 J mice to generate heterozygous F1 mice. The F1 mice were genotyped by PCR. The targeted F1 male and female mice were crossed to generate the F2 homozygous mice, which were genotyped (deletions of *HS7L*, *CBSf*, *CBSfg*, or *CBSg* were shown in Supplementary Figs. 4, 12, the other mouse lines were genotyped previously[49]) and used for downstream experiments. The wild-type F2 littermates were used as controls.

## ATAC-seq

ATAC-seq was performed following the omni-ATAC procedure[64] with some modifications. Briefly, the neocortex was microdissected and digested with 0.00625% collagenase followed by filtration through a 100 μm cell strainer to generate single-cell suspensions. The suspended cells were washed twice with PBS solution and then counted. $10^5$ cells were aliquoted and resuspended with 50 μl of the resuspension buffer (RSB) supplemented with 0.1% NP40, 0.1% Tween-20, and 0.01% digitonin followed by incubation on ice for 3 min for lysis. The nuclei were then washed with 1 ml of RSB supplemented with only 0.1% Tween-20 and resuspended in 50 μl of transposition mix, which includes 25 μl of 2× TD buffer, 3 μl transposase, 16 μl PBS, 0.5 μl 1% digitonin, 0.5 μl 10% Tween-20 and 5 μl $H_2O$, followed by incubation at 37 °C for 30 min in a shaker at 1000 rpm. Transposition products were immediately cleaned up with 1× Ampure XP Beads (Beckman, A63881), and the purified DNA was used for library construction by the TruePrep DNA Library Prep Kit V2 for Illumina (Vazyme Biotech, TD502). The ATAC-seq libraries were pooled and sequenced at the 2 × 150 bp mode using an Illumina NovaSeq 6000 platform.

## RNA-seq

Total RNA was extracted from cultured cells or the microdissected mouse neocortex using the Trizol reagent (ThermoFisher, 15596026). For each sample, 1 μg of total RNA was used to enrich mRNA using Poly(A) Magnetic Isolation Module (NEB, E7600). The mRNA was purified twice, eluted into the fragmentation buffer with random primers for fragmentation at 94 °C for 15 min, and reverse-transcribed into the first strand cDNA. The second strand was then synthesized by the second-strand DNA polymerase. The generated double-stranded cDNA was purified with Ampure XP beads (Beckman, A63881), end repaired, 3'-adenylated, and 5'-phosphorylated. The Illumina sequencing adapters were then ligated to both ends of cDNA and excess adapters were removed using Ampure XP beads (Beckman). The purified cDNA was amplified by PCR with barcoding to generate RNA-seq libraries. Pooled multiplexed libraries were sequenced at the 2 × 150 bp mode on an Illumina NovaSeq 6000 platform. All experiments were performed with at least two biological replicates.

Fetal sex is determined based on the expression of *Xist* and *Ddx3y*. For expression mapping at different embryonic stages during cortical development, mouse neocortices were isolated from both males (*Xist*⁺, *Ddx3y*⁻) and females (*Xist*⁻, *Ddx3y*⁺). As we did not observe sex-specific differences in *cPcdh* expression at any developmental stages (Supplementary Fig. 2f, g), we did not discriminate between male and female mouse neocortices for all the following experiments.

## ChIP-seq

Mouse neocortical tissues was microdissected and dissociated into single cells using 0.00625% collagenase. $2 \times 10^6$ of mouse neocortical or

cultured human cells were aliquoted and cross-linked using 1% formaldehyde in PBS containing 10% FBS for 10 min at room temperature followed by two washes with PBS containing proteinase inhibitors (Roche, 04693132001). Cells were then resuspended and lysed twice with 1 ml of prechilled lysis buffer containing proteinase inhibitors (Roche) for 10 min with slow rotations. For cortical cells, additional SDS was added to the lysis solution to adjust the final SDS concentration to 0.4% for better sonication. Cells were then sonicated at 25% power for a train of 15 s ON and 30 s OFF for 20 cycles with the Bioruptor system to break the genomic DNA into 0.1–10 kb fragments. After sonication, the SDS-free lysis buffer was added to cortical samples to dilute the SDS concentration to 0.1% to allow the binding of antibodies to proteins. After removal of the insoluble debris by centrifuging at 14,000 g for 10 min, the lysate was pre-cleaned with protein-A agarose beads (Millipore, 16–157) for 1 h with slow rotations to remove non-specific binding and then incubated with specific antibody (Supplementary Data 6) overnight with slow rotations at 4 °C. The antibody-protein-DNA complex was isolated by adding protein A-agarose beads and incubated for 4 h with slow rotations followed by sequential washes with the low-salt buffer, high-salt buffer, salt-free buffer, LiCl buffer, and TE buffer. The beads were eluted in the elution buffer (0.1 M $NaHCO_3$, 1% SDS) and de-cross-linked by proteinase K at 65 °C overnight. The DNA was finally purified and used for the construction of DNA library using the VAHTS Universal DNA Library Prep Kit for Illumina V3 (Vazyme Biotech, ND607). ChIP-seq libraries were sequenced at the 2 × 150 bp mode on an Illumina NovaSeq 6000 platform.

## QHR-4C

For QHR-4C[49] experiments, mouse neocortices were microdissected and dissociated with 0.00625% collagenase to obtain single cells. Mouse neocortical or cultured human cells were cross-linked in 2% formaldehyde for 10 min at room temperature and quenched by adding excess pre-chilled glycine. The fixed cells were incubated on ice for 5 min, and permeabilized twice with the permeabilization buffer (50 mM Tris-HCl pH 7.5, 150 mM NaCl, 5 mM EDTA, 0.5% NP-40, 1% Triton X-100, and protease inhibitors), with slow rotations for 10 min at 4 °C. The permeabilized cells were then digested with *Dpn* II for 16 h at 37 °C while shaking at 900 rpm followed by incubation at 65 °C for 20 min to inactive *Dpn* II. Proximity ligation was then performed for 16 h at 16 °C by adding T4 DNA ligase in 1x T4 ligation buffer. The ligated product was de-cross-linked by proteinase K at 65 °C overnight, treated with RNase A at 37 °C for 45 min, and purified with phenol-chloroform followed by ethanol precipitation. DNA was sonicated to 0.2–1 kb fragments using the Bioruptor system at 33% power for a train of 15 s ON and 30 s OFF for four cycles.

The anchor fragments were linear-amplified using a specific 5' biotin-tagged primer (Supplementary Data 5) with 120 cycles (95 °C 20 s for denaturing, 60 °C 20 s for annealing, and 72 °C 90 s for elongation). The amplified products were finally denatured at 95 °C for 5 min and then immediately chilled on ice to prevent reannealing. The biotin-tagged single-stranded DNA (ssDNA) was pulled down with Streptavidin Magnetic beads (ThermoFisher, 65001) and ligated with adapters which were generated by annealing two partially complementary single-stranded oligonucleotides (Adapter-U and Adapter-L, Supplementary Data 5) by T4 DNA ligase in 1x T4 ligation buffer containing PEG8000 at 16 °C for 12 h. The on-bead ligation products were washed twice with the B/W buffer (5 mM Tris-HCl pH 7.5, 1 M NaCl, and 0.5 mM EDTA) to remove excess adapters and used as the template for PCR amplification of the 4C libraries with barcoded anchor-specific P5-forward primers and indexed P7-reverse primers (Supplementary Data 5). Multiplexed libraries from the same anchor with different combinations of barcodes and indexes were pooled for purification with a PCR purification kit (Qiagen, 11732668001). 4C libraries were sequenced at the 2 × 150 bp mode on an Illumina NovaSeq 6000 platform.

## In situ Hi-C

For in situ Hi-C[8] experiments, $5 \times 10^6$ cells were cross-linked with formaldehyde and incubated in a lysis buffer (10 mM Tris-HCl pH8.0, 10 mM NaCl, 0.2% NP-40) to obtain nuclei. Nuclei were permeabilized with SDS, quenched by adding Triton X-100, and digested with *Mbo* II(NEB, R0147M) overnight at 37 °C, followed by heat inactivation at 62 °C. The ends of restriction fragments were filled in with biotin-14-dATP (Thermo, 19524016), dCTP, dTTP, and dGTP using the Klenow fragment of DNA polymerase I (NEB, M0210L), and ligated using T4 DNA ligase (NEB, M0202S). After ligation, the nuclei were de-cross-linked overnight at 68 °C. The ligated DNA was precipitated with ethanol and fragmented by sonication. 300-500 bp DNA fragments were selected using AMPure XP beads (Beckman, A63881). The biotinylated DNA was pulled down with M-280 Streptavidin beads (Thermo, 11206D), washed, and used for library construction on bead. The DNA was end-repaired, tailed with dA using Klenow exo⁻ (NEB, M0212S), and ligated with Illumina U-type adapters, followed by adapter linearization with a USER enzyme (NEB, M5505S). The obtained DNA was used as templates for PCR amplification with 10-12 cycles. PCR products were purified with AMPure XP beads (Beckman, A63881) and sequenced at the $2 \times 150$ bp mode on an Illumina Nova-Seq 6000 platform.

## High-throughput sequencing data analysis

RNA-seq reads were aligned against the mouse (GRCm38) or human (GRCh38) genome using Hisat2[65] (version 2.0.4) to generate the sequence alignment map (SAM) files. SAM files were then converted into the binary versions (BAM files) using Samtools (version 1.15.1). BAM files were used as input for Cufflinks[66] (version 2.1.1) to calculate FPKM values.

ChIP-seq and ATAC-seq reads were aligned to the mouse (mm9) or human (hg19) genome using Bowtie2 (version 2.3.5) to generate SAM files, which were converted into BAM files using Samtools (version 1.15.1). BAM files were indexed and converted into BedGraph files.

For 4 C data, P7 reads were initially demultiplexed based on the unique barcode-index combinations. The anchor primer sequences were trimmed and PCR duplicate reads were removed using FastUniq (version 1.1) of the UNIX system. The obtained unique reads were aligned against the mouse (mm9) or human (hg19) genome using Bowtie2 (version 2.3.5) to generate SAM files, which were converted into BAM files. The BAM files were used as input for r3Cseq (version 1.20) in the R package (3.3.3) to calculate the reads per million (RPM) values.

Hi-C reads were pre-processed with HiC-Pro (v3.0.0)[67], including reads alignment to the human hg19 reference genome, interaction matrix generation, and ICE normalization. 10 kb ICE normalized interaction matrix was generated. TADs and TAD boundaries were called at a 10-kb resolution using the directionality index (DI) method[51]. Insulation scores were calculated at a 10-kb resolution as previously described[68].

For genome-wide CBS patterns at TAD boundaries, we analyzed in situ HiC data. CBS position and orientation at TAD boundaries were determined by FIMO. We refer to a CTCF motif as "forward" (>) when it is present on the forward strand of the chromosome and 'reverse' (<) when it is on the reverse strand. We defined TAD boundary regions as ±50 kb from the center of each boundary. TAD boundary regions were intersected with CTCF motifs using "bedtools intersect" to identify CBS numbers and orientations for each TAD boundary.

## Reporting summary

Further information on research design is available in the Nature Portfolio Reporting Summary linked to this article.

## Data availability

The raw high-throughput sequencing data and processed data generated in this study have been deposited in the GEO database under accession code GSE210817. Source data are provided with this paper.

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

## Acknowledgements

We are grateful for technical supports from Jingwei Li, Leyang Wang, and Mo Zhang. This work was supported by grants from the National Natural Science Foundation of China (32330016), the National Key R&D Program of China (2022YFC3400200), and the Science and Technology Commission of Shanghai Municipality (21DZ2210200).

## Author contributions

Q.W. conceived the research. X.G. and K.H. performed experiments. H.H. and X.G. analyzed data. Z.W. and W.X. supervised the mouse experiments. X.G., H.H., and Q.W. wrote the manuscript with inputs from all authors.

## Competing interests

The authors declare no competing interests
