## [Peer Review File · Nature Communications]

Outward-oriented sites within clustered CTCF boundaries are key for intra-TAD chromatin interactions and gene regulationREVIEWER COMMENTS

Reviewer #1 (Remarks to the Author):

The authors chose the cPcdh locus as a model to study epigenetic regulation and 3D genome organization principles. Through deletion each CBS of candidate enhancers, they found a precise pattern of regulation of Pcdh β and Pcdhy gene expression by clustered CTCF sites within TAD boundaries. Deleting an outward-oriented CBSf within clustered CTCF sites reduced its interaction with Pcdh β /Pcdhy and their expression. They also studied the HOXD locus and found that deleting outward-oriented CBS3/5 at the C-DOM boundary reduced HOXD13's interaction with LNPk/GT2 and expression. This study provides novel insights into the complex mechanisms about the crucial role of clustered CTCF sites within TAD boundaries and highlights the importance of outward-oriented CBS elements in regulating intraTAD chromatin interactions and gene expression. However, the manuscript would benefit from a more comprehensive discussion of relevant theoretical frameworks and experimental evidence.

1. More background on loop extrusion and CBS element orientation at TAD boundaries and loop formation could be provided in the introduction. This would give readers a better theoretical framework to understand the role of CBS elements and their orientation in existing theories.

2. The authors demonstrated that HS5-1bL acts as a specific insulator only to insulate Pcdhy3. Does HS5-1bL also function as a proximal enhancer, or does it appear to function only as insulators among clustered CTCF sites to balance the fine regulation of cPcdh by downstream enhancers?

3. The discussion of HOXD locus could be extended to the overall structure of C-DOM and its effect on genes inside TAD. Does deletion of outward-oriented CBS affect the expression of genes inside C-DOM?

4. The authors thoroughly analyzed the role of outward-oriented CBS elements in boundary structure and gene regulation at two model loci. Although the authors noted in the conclusion that the generality of this finding needs further verification, statistical methods could be used to determine whether outward-oriented CBS elements are commonly present at clustered CTCF sites or TAD boundaries across the whole genome.

5. The authors propose an insightful model for how outward-oriented CBS elements and clustered CTCF sites control gene expression through regulating intra-TAD and inter-TAD interactions. However, the description of model figure is superficial. Furthermore, this model could be discussed in depth. For example, does the proposed model align with the loop extrusion model for TAD formation and enhancer-promoter contacts? How do outward-oriented CBS elements functionally differ from inward-oriented CBS elements at TAD boundaries?

Reviewer #2 (Remarks to the Author):

The function of CTCF sites in chromatin interaction at PCDH and HOXD regions have been extensively studied, including several studies of the authors. In this manuscript, the authors found outward-oriented CBS elements within clustered CBSs are also required to maintain cohesin extrusion chromatin loop by using CRISPR-deletion models of outward-oriented CBS elements at PCDH and HOXD regions. This is interesting but not surprising, because the previous CBS orientation study did not exclude the possibility of chromatin loop formation at CTCF sites with non-convergent CBSs.

1. The authors claimed HS5-1bL is a Pcdhy3-specific insulator based on 4C results (Fig. 2). This conclusion should be confirmed by Hi-C data. Even it is confirmed, this finding is not very relevant to the main point of this study.

2. The authors provided some histone modification data of cPcdhs regions (Fig. 3), but these data are all descriptive and not related to other parts of this manuscript.

3. Since CBS f and g are very close and in the same CTCF peak, it is hard to distinguish the effect of f and g. The authors claimed that CBSf-deletion abolished CTCF/Rad21 binding at f but not g

(line 207~210). However, compared with WT, CTCF/Rad21 binding is also dramatically disrupted at g site (Fig .4b). To confirm this is CBSf site specific effect, CBSg-deletion is needed for all downstream analyses.

4. In Fig. 4d, Why CBSf-deletion results an increase of chromatin interaction between Pcdh β/γ genes and CBSH region? There is no mechanistic explanation in the manuscript for the major finding. Is it because Cohesin bypass CBSf site and stop at CBSH site? But from the ChIP-Seq data of Fig4a, CTCF/Rad21 binding is not changed at CBSH. Actually, CTCF/Rad21 binding affinity is very low at CBSH site both before and after CBSf deletion, indicating CBSH site is not a cohesin loop anchor.

5. CRISPR-deletion at two genomic regions could not generalize the finding. To assess the role of these CBSs in chromatin looping, it is needed to systematically characterize the outward-oriented CBS elements by CRISPR screening and compare them with inward-oriented CBS elements.

6. The biological significance of this study is not clear. Is any of the outward-oriented CBS elements deleted in this study mutated in any human disease or dysregulated in development? Since the novelty of the finding is limited, the biological relevance is important to justify the necessity of this study.

7. In Fig1. e-k and Fig4. c, how to understand that some isoforms ($\gamma a7$, $\gamma a11$, $\gamma a12$) show increased gene expression after the deletion of putative enhancers?

Point-by-point response to the reviewers' comments

Thank you very much for your time and effort in reviewing our manuscript. We really appreciate the insightful comments and invaluable critiques. In response, we have performed additional experiments of CBSg deletion, chromosome conformation capture, RNA-seq, and histone modifications for the protocadherin cluster. The data are shown in the new Supplementary Figs 12e-g, 13c, 14-17, and 19. In addition, we have performed additional experiments of chromosome conformation capture, RNA-seq, and histone modifications experiments with single-cell clones of single CBS deletions of CBS3 or CBS5 for the HoxD cluster. The data are shown in the new Supplementary Figs 20e,f, 21c,d, and 22. All other figures and tables are unchanged except specified.

Point-by-point response to the reviewers' comments are shown as following (original comments are in *Italic* and our response are in **Bold**).

Reviewer #1 (Remarks to the Author)

The authors chose the cPcdh locus as a model to study epigenetic regulation and 3D genome organization principles. Through deletion each CBS of candidate enhancers, they found a precise pattern of regulation of Pcdh β and Pcdhy gene expression by clustered CTCF sites within TAD boundaries. Deleting an outward-oriented CBSf within clustered CTCF sites reduced its interaction with Pcdh β /Pcdhy and their expression. They also studied the HOXD locus and found that deleting outward-oriented CBS3/5 at the C-DOM boundary reduced HOXD13's interaction with LNPk/GT2 and expression. This study provides novel insights into the complex mechanisms about the crucial role of clustered CTCF sites within TAD boundaries and highlights the importance of outward-oriented CBS elements in regulating intraTAD chromatin interactions and gene expression. However, the manuscript would benefit from a more comprehensive discussion of relevant theoretical frameworks and

experimental evidence.

Thank you very much for the positive comments and constructive suggestions. To further improve the manuscript as suggested, we have performed additional experiments and made comprehensive discussion of our finding on the outward CBS elements within the relevant “loop extrusion” frameworks.

1. More background on loop extrusion and CBS element orientation at TAD boundaries and loop formation could be provided in the introduction. This would give readers a better theoretical framework to understand the role of CBS elements and their orientation in existing theories.

Thank you for the suggestion. We have added one sentence in the Introduction at the lines 45 to 48 (L45-L48) on Page 3 (P3) to provide additional explanation on loop extrusion: “Specifically, continuous ATP-driven active loop extrusions lead to gradually larger-sized chromatin loops which are stabilized by pairs of convergent or opposite CBS elements.”

2. The authors demonstrated that HS5-1bL acts as a specific insulator only to insulate Pcdh γ 3. Does HS5-1bL also function as a proximal enhancer, or does it appear to function only as insulators among clustered CTCF sites to balance the fine regulation of cPcdh by downstream enhancers?

Thanks for your kind question. Pcdh γ 5 and Pcdh γ 4 are the most proximal genes to HS5-1bL (Fig. 2a), but neither of them is affected in their expression by HS5-1bL deletion (Fig. 2e). These data suggest that HS5-1bL may not function as a proximal enhancer. It may function as Pcdh γ 3-specific insulator among clustered CTCF sites to balance the fine regulation of cPcdh by downstream enhancers.

3. The discussion of *HOXD* locus could be extended to the overall structure of C-DOM and its effect on genes inside TAD. Does deletion of outward-oriented CBS affect the expression of genes inside C-DOM?

Thanks for the constructive suggestion that will significantly improve the clarity of our manuscript. The C-DOM also contains one other gene of *LNP*K but it does not associate with any CBS element and its regulation was not altered by deletion of outward CBS elements. We have added the following text at L269-275 at P11: “However, different from the *Pcdhβγ* genes which are associated with CBS elements, there is no CBS element associated with the *LNP*K gene within C-DOM and its expression is not altered upon deletion of outward-oriented CBS elements (Fig. 5g). Finally, single deletions of the CBS3 or CBS5 element demonstrated a cooperation role between the two outward-oriented CBS elements in *HOXD13* gene regulation (Supplementary Fig. 22).”

4. The authors thoroughly analyzed the role of outward-oriented CBS elements in boundary structure and gene regulation at two model loci. Although the authors noted in the conclusion that the generality of this finding needs further verification, statistical methods could be used to determine whether outward-oriented CBS elements are commonly present at clustered CTCF sites or TAD boundaries across the whole genome.

Thanks for the constructive suggestion. We have analyzed the CBS orientation patterns of TAD boundaries across the whole genome in human HEC-1-B cells and found majorities have outward CBS elements. We have added the following text at L370-372 on P14: “We analyzed genome-wide CBS patterns at TAD boundaries and found that 85.7% have outward CBS elements (Supplementary Data 2 and 3).” These data show that outward-oriented CBS elements are commonly present at TAD boundaries across the whole genome.

5. The authors propose an insightful model for how outward-oriented CBS elements and clustered CTCF sites control gene expression through regulating intra-TAD and inter-TAD interactions. However, the description of model figure is superficial. Furthermore, this model could be discussed in depth. For example, does the proposed model align with the loop extrusion model for TAD formation and enhancer-promoter contacts? How do outward-oriented CBS elements functionally differ from inward-oriented CBS elements at TAD boundaries?

Thanks for the constructive suggestion. As suggested, we have expanded the description of our model at L296-300 on P12: “Unlike inward-oriented CBS elements, which directly make intraTAD contacts, such as between enhancers and promoters, the outward-oriented CBS elements may block extruding cohesin sliding from the outside of the TAD to prevent aberrant interTAD chromatin interactions (Fig. 5h).” Our model aligns well with the loop extrusion model for TAD formation and enhancer-promoter contacts and suggests both outward and inward CBS elements at clustered CTCF TAD boundaries are important for chromatin interaction and gene regulation.

Reviewer #2 (Remarks to the Author)

The function of CTCF sites in chromatin interaction at PCDH and HOXD regions have been extensively studied, including several studies of the authors. In this manuscript, the authors found outward-oriented CBS elements within clustered CBSs are also required to maintain cohesin extrusion chromatin loop by using CRISPR-deletion models of outward-oriented CBS elements at PCDH and HOXD regions. This is interesting but not surprising, because the previous CBS orientation study did not exclude the possibility of chromatin loop formation at CTCF sites with non-convergent CBSs.

The loops and TADs are thought to be formed by loop extrusion and its

asymmetric blocking by CTCF at convergent CBS elements. Thus, the TAD flanking boundaries are normally demarcated by inward-oriented CBS elements, which can directly contribute to the intra-TAD CTCF/cohesin-mediated enhancer-promoter interactions as shown previously. However, outward-oriented CBSs are frequently located within clustered CTCF boundaries (Supplementary Data 2 and 3). In this manuscript, we focused on the role of the outward-oriented boundary CBSs instead of loop formation at non-convergent CBSs or tandem CBS elements (tandem left or tandem right CBS pairs). We found outward-oriented CBS elements within clustered CTCF boundaries may play a key role in blocking the cohesin sliding from the outside of TAD to prevent aberrant interTAD promoter-anchored chromatin interactions and to maintain proper intraTAD chromatin contact and gene regulation.

1. The authors claimed *HS5-1bL* is a *Pcdh γ 3*-specific insulator based on 4C results (Fig. 2). This conclusion should be confirmed by Hi-C data. Even it is confirmed, this finding is not very relevant to the main point of this study.

We conclude *HS5-1bL* as a *Pcdh γ 3*-specific insulator based on both 4C results (Fig. 2) and RNA-seq data (Fig 1i and Supplementary Fig4e). RNA-seq data showed that, within the *Pcdh $\beta\gamma$* TAD or the whole *Pcdh* superTAD, only *Pcdh γ 3* is significantly increased in expression by *HS5-1bL* deletion while none of other genes is affected. 4C results showed that, the interaction between *Pcdh γ 3* and enhancers of *HS18-20* is significantly increased by *HS5-1bL* deletion.

For specific loci, 4C is superior than Hi-C although they are both based on 3C. 4C allows identification of genome-wide interactors with one element (the “anchor”) at higher resolution than Hi-C (de Wit and de Laat. *Genes Dev.* 26(1):11-24, 2012). In 4C, four-base-recognizing enzyme, such as *DpnII* used in this manuscript, produces restriction fragments with

averaged size of 256 bp. While in Hi-C, it is difficult and costly to sequence at sufficient depth to provide enough resolution for interrogating specific contacts between promoters and distal regulatory elements. Most of the published mammalian Hi-C interactome maps are binned at 10-100 kb resolution. *Pcdh γ 3* and *HS18-20* are too close (~70 kb) to allow interrogation of the specific contacts between them by Hi-C (Response Fig. 1). We thus used 4C to profiles their interactions.

Response Fig. 1 | Hi-C map around *HS5-1bL* at 10 kb resolution.

In order to clarify the function of outward-oriented CTCF sites within the TAD boundary, we made great efforts on how members of the *Pcdh $\beta\gamma$* clusters are regulated within the *Pcdh $\beta\gamma$* TAD. We deleted each of the candidate enhancers to assess their contribution to *Pcdh $\beta\gamma$* gene expression. We surprisingly found deletion of *HS5-1bL* only causes an increase in *Pcdh γ 3* expression. Given the unique role of *Pcdh γ 3* in the brain development, we found that *HS5-1bL* is a *Pcdh γ 3*-specific insulator for its specific regulation. We agree that this finding is not very relevant to the main point of this study and will be happy to remove it.

2. *The authors provided some histone modification data of cPcdhs regions (Fig. 3), but these data are all descriptive and not related to other parts of this manuscript.*

We agree that histone modification data are descriptive and not related to other parts of this manuscript. To show their relevance to gene regulation in the CBSf and CBSfg deletion mice, we performed additional ChIP-seq experiments with a series of specific antibodies against H3K9me3, H3K9me2, H3K36me3, H3K4me3, or H3K9ac, and found decrease of repressive marks of H3K9me3 and H3K9me2, but not active marks of H3K36me3, H3K4me3, and H3K9ac. We added the following text at L227-229 on P9: “In addition, we profiled the histone modifications and found decrease in enrichments of repressive marks, H3K9me3 and H3K9me2, but not of active marks, H3K36me3, H3K4me3 and H3K9ac (Supplementary Figs. 14-17).”

3. *Since CBS f and g are very close and in the same CTCF peak, it is hard to distinguish the effect of f and g. The authors claimed that CBSf-deletion abolished CTCF/Rad21 binding at f but not g (line 207~210). However, compared with WT, CTCF/Rad21 binding is also dramatically disrupted at g site (Fig .4b). To confirm this is CBSf site specific effect, CBSg-deletion is needed for all downstream analyses.*

Thank you for the insightful suggestion. We have generated the CBSg-deletion and performed CTCF and Rad21 ChIP-seq as well as RNA-seq and 4C experiments. The data are shown in the new Supplementary Fig. 19. Compared with original data of CBSf-deletion (Fig. 4) and CBSfg-deletion (Supplementary Fig. 18), the effect of CBSg-deletion is weaker than CBSf-deletion or CBSfg-deletion although the CTCF binding at the CBSf and CBSg elements appears to be cooperative. We have added the following text at L232-234 at P10: “As a control, single deletion of CBSg

results in a less decrease in *Pcdh $\beta\gamma$* expression (Supplementary Figs. 12f,g,19).”

4. In Fig. 4d, Why CBSf-deletion results an increase of chromatin interaction between *Pcdh $\beta\gamma$* genes and CBS β region? There is no mechanistic explanation in the manuscript for the major finding. Is it because Cohesin bypass CBSf site and stop at CBS β site? But from the ChIP-Seq data of Fig4a, CTCF/Rad21 binding is not changed at CBS β . Actually, CTCF/Rad21 binding affinity is very low at CBS β site both before and after CBSf deletion, indicating CBS β site is not a cohesin loop anchor.

As suggested, we have added a mechanistic explanation for increased chromatin interaction between *Pcdh $\beta\gamma$* genes and CBS β region in the text at L332-337 on P13 as following: “In the case of clustered CTCF TAD boundary of *Pcdh $\beta\gamma$* , the extruding cohesin complexes from the downstream TAD cannot bypass the forward CBSf element. However, upon deletion of the outward CBSf element, these upward-extruding cohesin complexes bypass the entire TAD boundary and eventually anchored at variable promoters, leading to an increase of chromatin interactions between *Pcdh $\beta\gamma$* genes and CBS β region.”

5. CRISPR-deletion at two genomic regions could not generalize the finding. To assess the role of these CBSs in chromatin looping, it is needed to systematically characterize the outward-oriented CBS elements by CRISPR screening and compare them with inward-oriented CBS elements.

Thank you for providing a great idea of systematically characterizing the outward-oriented CBS elements by CRISPR screening. We have systematically analyzed the numbers and orientations of the TAD boundary CBS elements across the human genome and added the results in Supplementary Data 2 and 3. However, it is beyond the scope of the current manuscript and a big challenge to perform the CRISPR screening genome wide in the short revision timeframe. We have toned down the

description by adding the following sentence in the text at L372 and L373 on P14: “However, whether it could generalize to the entire genome awaits further studies.”

6. *The biological significance of this study is not clear. Is any of the outward-oriented CBS elements deleted in this study mutated in any human disease or dysregulated in development? Since the novelty of the finding is limited, the biological relevance is important to justify the necessity of this study.*

Thanks for the suggestion to improve the biological significance of our manuscript. We failed to find mutations of the outward-oriented CBS elements deleted in our study in human GWAS databases of GWAS Catalog and GWAS Central. However, we have found many human pathogenic CNVs associated with these CBS elements from the ClinGen and DECIPHER databases. We have listed them in Supplementary Data 1 and added the following text at L291 and L292 on P12: “which are associated with dozens of human pathogenic copy number variations (CNVs) (Supplementary Data 1).”

7. *In Fig1. e-k and Fig4. c, how to understand that some isoforms ($\gamma a7$, $\gamma a11$, $\gamma a12$) show increased gene expression after the deletion of putative enhancers?*

The expression levels of *Pcdh $\gamma a7$* and *$\gamma a11$* was increased upon deletions of *HS19-20* (Δ *HS19-20*, Δ *HS18-20*, and Δ *HS5-1bL-HS20*) (Fig. 1f,h,k). Consistently, the activation of *Pcdh $\gamma a7$* by deletion of *HS19-20*-containing region was also observed in the mouse whole brain at E18.5 as detected by qPCR (Yokota et al. JBC 286: 31885-31895, 2011). These data suggest a repressive role of *HS19-20* in *Pcdh $\gamma a7$* and *$\gamma a11$* transcription. The increased expression levels of *Pcdh $\gamma a12$* in Δ *HS5-1bL-HS18* mice (Fig. 1g) might be contributed by the region between *HS5-1bL* and *HS18*, since no alteration of expression levels was observed in Δ *HS5-1bL* or Δ *HS18* mice

(Fig. 1i,j).

REVIEWER COMMENTS

Reviewer #1 (Remarks to the Author):

I have no further questions.

Reviewer #2 (Remarks to the Author):

The authors have made efforts to address many of my questions through additional experiments and text revisions. However, one major concern remains regarding the mechanistic explanation for the increased chromatin interactions upon CBSf deletion.

The added statement proposes that deletion of CBSf enables upward extruding cohesin complexes to bypass the TAD boundary and anchor at promoters, increasing Pcdh-CBS_h contacts. To support this model, it is necessary to compare Rad21 binding on both sides of CBSf in wildtype and deleted cells, like the 4C ratio analysis in Figure 4e. Figure 4b only shows a small window of Rad21 binding, which is insufficient compared to the 4C ratio analysis. If the model is correct, I would expect to see increased Rad21 binding downstream of the deleted CBSf in knockout compared to wildtype cells, even if binding at CBS_{g/h} is unchanged. Alternatively, if Rad21 binding distribution is unaltered upon CBSf deletion, further explanation would be needed to reconcile the 4C data with the proposed model. Therefore, the authors must include further Rad21 binding analysis of CBSf deleted cells to directly test the model, or adjust the model if Rad21 data does not match expectations.

Point-by-point response to the reviewers' comments

Dear reviewers,

Thanks again for your time and effort in reviewing our manuscript. We really appreciate the insightful advices that help to improve academic rigor of our article. In response, we have performed a Rad21 binding analysis of CBSf-deleted cells and adjusted the working model accordingly. The Rad21 binding analysis data of CBSf-deleted mice are shown in the new Supplementary Fig. 12b. The adjusted model is shown in Fig 5h.

Point-by-point response to the reviewers' comments are shown as following (original comments are in *Italic* and our response are in **Bold**).

Reviewer #1 (Remarks to the Author)

I have no further questions.

Thank you again for the positive comments and constructive suggestions.

Reviewer #2 (Remarks to the Author)

The authors have made efforts to address many of my questions through additional experiments and text revisions. However, one major concern remains regarding the mechanistic explanation for the increased chromatin interactions upon CBSf deletion.

The added statement proposes that deletion of CBSf enables upward extruding cohesin complexes to bypass the TAD boundary and anchor at promoters, increasing Pcdh-CBSH contacts. To support this model, it is necessary to compare Rad21 binding on both sides of CBSf in wildtype and deleted cells, like the 4C ratio analysis in Figure 4e. Figure 4b only shows a small window of Rad21 binding, which is insufficient compared to the 4C ratio analysis. If the model is correct, I would expect to see increased Rad21 binding downstream of the deleted CBSf in knockout compared to wildtype cells, even if binding at CBSg/h is unchanged. Alternatively, if Rad21 binding distribution is unaltered upon CBSf deletion, further explanation would be needed to reconcile the 4C

data with the proposed model. Therefore, the authors must include further Rad21 binding analysis of CBSf deleted cells to directly test the model, or adjust the model if Rad21 data does not match expectations.

Thank you for raising your concerns regarding the mechanistic explanation for the increased chromatin interactions upon *CBSf* deletion. As you suggested, we have compared Rad21 binding on both sides of *CBSf* in wildtype and *CBSf*-deleted mice (Supplementary Fig. 12b). However, we did not observe increased Rad21 binding downstream of the deleted *CBSf* in knockout compared to wildtype cells. The description has been added in the Results at the lines 213 to 214 on Page 9: “**CTCF and cohesin binding at both sides of *CBSf* is not altered by *CBSf* deletion (Supplementary Fig. 12b).**” As suggested by Reviewer2, we have modified the working model in Fig. 5h accordingly. The cohesin complexes are dynamically loaded onto DNA and extrude a loop continuously until reaching convergent CBS elements, such as the forward-oriented *CBSf* and the reverse-oriented *CBS_h*. Upon *CBSf* deletion, these extruding cohesin complexes pass through all the upstream reverse-oriented CTCF sites within the TAD boundary and anchor at the long-distance forward-oriented promoter sites, explaining increased long-distance chromatin interactions, but not cohesin enrichment, with the downstream *CBS_h* region.